# ADAMTS-13: A Prognostic Biomarker for Portal Vein Thrombosis in Japanese Patients with Liver Cirrhosis

**DOI:** 10.3390/ijms25052678

**Published:** 2024-02-26

**Authors:** Junya Suzuki, Tadashi Namisaki, Hiroaki Takya, Kosuke Kaji, Norihisa Nishimura, Akihiko Shibamoto, Shohei Asada, Takahiro Kubo, Satoshi Iwai, Fumimasa Tomooka, Soichi Takeda, Aritoshi Koizumi, Misako Tanaka, Takuya Matsuda, Takashi Inoue, Yuki Fujimoto, Yuki Tsuji, Yukihisa Fujinaga, Shinya Sato, Koh Kitagawa, Hideto Kawaratani, Takemi Akahane, Akira Mitoro, Masanori Matsumoto, Kiyoshi Asada, Hitoshi Yoshiji

**Affiliations:** 1Department of Gastroenterology, Nara Medical University, 840 Shijo-cho, Kashihara 634-8522, Nara, Japan; suzukij@naramed-u.ac.jp (J.S.); htky@naramed-u.ac.jp (H.T.); kajik@naramed-u.ac.jp (K.K.); a-shibamoto@naramed-u.ac.jp (A.S.); asahei@naramed-u.ac.jp (S.A.); kubotaka@naramed-u.ac.jp (T.K.); satoshi181@naramed-u.ac.jp (S.I.); tomooka@naramed-u.ac.jp (F.T.); souitit@naramed-u.ac.jp (S.T.); yuring0309@naramed-u.ac.jp (A.K.); mtanaka@naramed-u.ac.jp (M.T.); takuya@naramed-u.ac.jp (T.M.); yukifuji@naramed-u.ac.jp (Y.F.); tsujih@naramed-u.ac.jp (Y.T.); fujinaga@naramed-u.ac.jp (Y.F.); shinyasato@naramed-u.ac.jp (S.S.); kawara@naramed-u.ac.jp (H.K.); stakemi@naramed-u.ac.jp (T.A.); mitoroak@naramed-u.ac.jp (A.M.); yoshijih@naramed-u.ac.jp (H.Y.); 2Department of Evidence-Based Medicine, Nara Medical University, 840 Shijo-cho, Kashihara 634-8522, Nara, Japan; tkinoue0@naramed-u.ac.jp; 3Department of Hematology, Nara Medical University, 840 Shijo-cho, Kashihara 634-8522, Nara, Japan; mmatsumo@naramed-u.ac.jp; 4Clinical Research Center, Nara Medical University, Nara Medical University, 840 Shijo-cho, Kashihara 634-8522, Nara, Japan; kasada@naramed-u.ac.jp

**Keywords:** ADAMTS-13:AC, portal vein thrombosis, liver cirrhosis

## Abstract

Portal vein thrombosis (PVT), one of the most prevalent hepatic vascular conditions in patients with liver cirrhosis (LC), is associated with high mortality rates. An imbalance between a disintegrin-like metalloproteinase with thrombospondin type-1 motifs 13 (ADAMTS-13) enzyme and von Willebrand factor (VWF) is responsible for hypercoagulability, including spontaneous thrombus formation in blood vessels. Herein, we aimed to identify potential prognostic and diagnostic biomarkers in Japanese patients with LC and PVT. In total, 345 patients were divided into two groups: 40 patients who developed PVT (PVT group) and 305 who did not develop PVT (NPVT group). Among the 345 patients with LC, 81% (279/345) were deemed ineligible due to the presence of preventive comorbidities, active or recent malignancies, and organ dysfunction. The remaining 66 patients were divided into two groups: the PVT group (n = 33) and the NPVT group (n = 33). Plasma ADAMTS-13 activity (ADAMTS-13:AC) and the vWF antigen (VWF:Ag) were measured using enzyme-linked immunosorbent assays. Contrast-enhanced, three-dimensional helical computed tomography (CT) was used to detect and characterize PVT. ADAMTS-13:AC was significantly lower in the PVT group than in the NPVT group. No significant differences in plasma vWF:Ag or liver stiffness were observed between the two groups. ADAMTS-13:AC of <18.8 was an independent risk factor for PVT on multivariate analyses (odds ratio: 1.67, 95% confidence interval: 1.21–3.00, *p* < 0.002). The receiver operating characteristic analysis of ADAMTS-13:AC revealed an area under the curve of 0.913 in PVT detection. Patients with PVT having ADAMTS-13:AC ≥18.8 (n = 17) had higher albumin levels and better prognoses than those with ADAMTS-13:AC <18.8 (n = 16). No significant correlations of ADAMTS-13:AC levels with either fibrin degradation product or D-dimer levels were observed. ADAMTS-13:AC levels could be potential diagnostic and prognostic biomarkers for PVT in Japanese patients with LC.

## 1. Introduction

PVT is the most common thrombotic complication in patients with liver cirrhosis (LC) [1]. PVT is generally asymptomatic and is frequently detected during periodic imaging surveillance of hepatocellular carcinoma (HCC) and hospitalization due to portal hypertension-related complications, including esophageal variceal bleeding [2]. Given the uniqueness of the portal venous system, the increased risk of thrombosis in the portal system could be attributed to the splanchnic territory in LC [3]. PVT is more commonly diagnosed in its advanced stages [4], with its incidence increasing by up to 25% in liver transplantation candidates and 35% in patients with hepatocellular carcinoma (HCC) [5]. The prevalence of PVT in patients with LC varied widely in reported studies, ranging from 0.6% to 15.8% [6]. The incidence of PVT was reported as 1.6% at 1 year, 6% at 3 years, and 8.3% at 5 years in a prospective study of 369 patients with LC [7]. PVT can spontaneously disappear during a short follow-up period [8]. Low platelet counts, decreased portal blood flow velocity, and episodes of variceal hemorrhage are independently associated with a high risk of PVT in patients with LC [7]. D-dimer and P-selectin levels are good potential predictive markers for PVT in patients with cirrhotic portal hypertension who have undergone devascularization procedures [9]. However, the risk factors for PVT remain relatively unknown.

Cirrhosis-induced coagulopathy encompasses disturbances in procoagulant and anticoagulant reactions, rendering compensatory hemostatic regulatory mechanisms inefficient in patients with cirrhosis. A disintegrin-like metalloproteinase with thrombospondin type-1 motifs 13 (ADAMTS13) is a zinc-containing metalloproteinase that cleaves the multimeric von Willebrand factor (VWF) between Tyr1605 and Met1606 residues in the central A2 domain [10]. ADAMTS13 is expressed mainly in hepatic stellate cells (HSCs) adjacent to sinusoidal endothelial cells (SECs) [11]. VWF, a multimeric glycoprotein, is produced and secreted in vascular ECs of different sizes and is released into the plasma as unusually large VWF (ULvWF) multimers. The hemostatic and thrombogenic potential of vWF depends on the multimer size [12]. VWF stimulates platelet thrombus formation in a high-shear-stress environment, and ULvWF multimers are more likely to adhere to platelets to induce platelet adhesion and aggregation in the circulatory system [13]. ADAMTS-13 deficiency causes the local elevation of ULvWF multimers in sinusoids. This phenomenon may support platelet thrombus formation within the hepatic microvasculature and macrovasculature, including the portal vein [14]. Therefore, in LC, impaired ADAMTS13 production results in an elevated intra-sinusoidal presence of ULvWF multimers and plasma coagulation factors, followed by PVT.

ADAMTS-13 activity is independently associated with PVT in patients with LC [15]. The ADAMTS-13/VWF ratio has been identified as a biomarker that predicts the onset of PVT in patients with LC [16]. More recently, bacterial infections have been identified as a risk factor for PVT development in patients with LC [17]. PVT is considered to be a negative prognostic factor in patients with cirrhosis [18]. A low plasma ATIII level is considered a risk factor for mortality in patients with LC [19], since ATIII administration is beneficial as one of the essential therapies for patients with PVT for those with ATIII of <70% [20]. However, factors that affect the prognosis of patients with LC complicated with PVT remain largely unknown. In this study, we aimed to identify prognostic markers in Japanese patients with LC and PVT.

## 2. Results

### 2.1. Comparison of Clinical Characteristics

Table 1 summarizes the baseline clinical characteristics and laboratory data of all patients. Out of 66 patients with LC, 40 (60.6%) were male, and the mean age of all of our study participants was 70.0 ± 8.8 years. The mean Child–Pugh score (CPS) was 7.2 ±1.9, with 26 (39.4%), 28 (42.4%), and 12 (18.2%) patients classified into CP grades (CPG) A, B, and C, respectively. The mean observation period was 1033 ± 147 days. The PVT group showed a trend toward higher liver stiffness measurement compared to the NPVT group. The type of PVT included Yerdel grade 1 in 28 (84.8%) cases, grade 2 in 4 (12.1%) cases, and grade 3 in 1 (14.3%) case. There were no cases of Yerdel grade 4 PVT. No significant differences were found in fibrin degradation product (FDP) and D-dimer levels between the PVT and NPVT groups. Overall, the patients with PVT did not show a worse clinical status compared with patients without PVT.

### 2.2. Plasma ADAMTS-13:AC and vWF:Ag Levels

ADAMTS-13:AC levels were significantly lower in the PVT group than in the NPVT group (*p* < 0.001). No significant differences in vWF:Ag levels were observed between the two groups. ADAMTS-13:AC levels were significantly lower in patients with Child–Pugh (CP) class A, B, and C cirrhosis (all *p* < 0.001) (Figure 1), whereas there were no significant differences in vWF:Ag levels were observed between PVT and NPVT groups in terms of the CP classification. Univariate and multivariate analysis demonstrated that ADAMTS-13:AC <18.8% is a significant risk factor for PVT in patients with LC (OR: 1.56; 95% confidence [CI]: 1.10–2.34; *p* = 0.0052) (OR: 1.67; 95% confidence [CI]: 1.21–3.00; *p* = 0.002) (Table 2).

### 2.3. Predictors of PVT in Patients with LC

Patients with LC were divided into two categories according to optimal cutoff levels for PVT detection using ADAMTS-13:AC levels. Receiver operating characteristic analyses showed that the area under the ROC curve of ADAMTS-13:AC for PVT detection in patients with cirrhosis was 0.913 (Figure 2) and the cutoff with predictive value for the detection of PVT was <18.8% with 81.8% sensitivity, 97.0% specificity, 96.5% positive predictive values, and 84.2% negative predictive values (Table 3).

### 2.4. Kaplan–Meier Survival Curve Comparisons of Patients According to Optimal Cutoff Levels of ADAMTS-13:AC for PVT Diagnosis in LC

Patients with PVT were divided into two groups based on the mean average value of ADMSTS13 for the PVT group: low ADAMTS-13:AC levels < 18.8% (n = 16) and high ADAMTS-13:AC levels ≥ 18.8% (n = 17). Patients with high ADAMTS-13:AC levels had significantly higher albumin levels and significantly lower death rates than those with low ADAMTS-13 levels (Table 4). Kaplan–Meier curve analyses revealed that patients with high ADMTS13:AC levels had more favorable survival durations (*p* = 0.031) (Figure 3).

### 2.5. Correlations of Coagulation and Fibrinolysis Parameters with ADAMTS-13:AC and vWF:Ag

No significant correlations of ADAMTS-13:AC levels with either FDP or D-dimer levels were found (Figure 4). No significant correlations of vWF:Ag levels with either FDP or D-dimer levels were found.

### 2.6. ADAMTS-13:AC Levels for Different Etiologies

ADAMTS-13:AC levels were shown in the PVT and non-PVT groups. The number of patients was too small to compare ADMTS13 values among different etiologies (Figure 5).

## 3. Materials and Methods

### 3.1. Patients

In this retrospective study, we enrolled 345 Japanese patients with liver cirrhosis who received treatment at the Department of Gastroenterology in Nara Medical University Hospital between July 2014 and August 2021. A total of 345 patients were divided into two groups: 40 patients who developed PVT (the PVT group) and 305 who did not develop PVT (the NPVT group). ADAMTS13:AC and VWF:Ag levels could be affected in patients with hepatocellular carcinoma (HCC), extrahepatic malignancy, other thromboses, and anticoagulation therapy [21]. Contrast-enhanced computed tomography (CECT) was not performed in patients with severe renal dysfunction or bronchial asthma to assess PVT in patients with LC. Therefore, patients with hepatocellular carcinoma (HCC; n = 3) and anticoagulation therapy (n = 4) were excluded from the PVT group. Patients with HCC (n = 55), extrahepatic malignancy (n = 25), anticoagulation therapy (n = 37), severe renal dysfunction (n = 35), bronchial asthma (n = 17), and other thrombosis (n = 14) and without contrast-enhanced computed tomography (CT) (n = 89) were excluded from the NPVT group. No patients with congenital coagulation factor deficiencies or severe thrombocytopenia were included in this study. In total, 66 patients for the PVT (n = 33) and NPVT (n = 33) groups were included. LC was diagnosed based on relevant clinical and paraclinical data, including laboratory tests (e.g., Alb, bilirubin, and prothrombin time), medical imaging features, liver histology, and clinical complications (e.g., hepatic encephalopathy and ascites). All patients underwent liver stiffness measurement (LSM) using transient elastography (Fibro scan) on the day the blood tests were performed. Patients with the following potential complications were excluded from the current study cohort: previous or ongoing PVT; active HCC on imaging or previously treated HCC or extrahepatic malignant tumors; those receiving anticoagulants; and those with severe renal dysfunction, bronchial asthma, and other forms of thrombosis (including deep vein thrombosis in the leg and pulmonary embolism, ischemic stroke, and myocardial infarction). The etiological classification of liver cirrhosis (LC) was as follows: (1) viral hepatitis (caused by the hepatitis B virus (HBV); n = 7, 10.6% and hepatitis C virus (HCV); n = 25, 37.9%); (2) alcoholic-related liver disease [22] (n = 13, 19.7%); (3) metabolic dysfunction-associated steatohepatitis ((MASH); n = 7, 10.6%); and (4) others including autoimmune hepatitis (AIH), primary biliary cirrhosis, and idiopathic causes (n = 14, 21.2%). Viral hepatitis was diagnosed according to the generally accepted serological criteria, including positivity for the HBs antigen to diagnose HBV infection and positivity for the HCV-III antibody and HCV RNA. ALD is currently diagnosed based on a person’s history of alcohol consumption (alcohol intake of ≥60 g/day and ≥40 g/day for males and females), physical examination, and laboratory test results. Metabolic dysfunction-associated liver disease is diagnosed histopathologically. All patients with HBV underwent nucleic acid analog therapy during the observation periods. Before or during the observation periods, 14 patients with hepatitis C infection underwent treatment with interferon or direct-acting antivirals. The Child–Pugh classification was used to describe the clinical state of patients with cirrhosis and evaluate the severity of liver disease [23]. No patient developed HCC during the follow-up period. Contrast-enhanced three-dimensional (3D) helical CT was used in the detection and characterization of PVT. All patients with cirrhosis received dietary advice and verbal explanations from a nutritionist. Dietary protein intake showed no remarkable differences in individual patients. All 13 patients died of liver-related causes. The study protocol conformed to the ethical guidelines of the Declaration of Helsinki. The Medical Ethics Committee of Nara Medical University approved our study protocol (Nara-medi, 143-1232-9). All patients provided their written informed consent for blood samples to be collected before study enrollment.

### 3.2. Determination of ADAMTS-13:AC and vWF:Ag

ADAMTS-13 activity (ADAMTS-13:AC) and vWF antigen (VWF:Ag) levels were measured at the Nara Medical University Hospital. Blood samples were collected from patients at the time of admission or during regular outpatient treatment and were stored in plastic tubes containing 0.38% volume of sodium citrate. Platelet-poor plasma was prepared via centrifugation at 3000× *g* and 4 °C for 15 min and was stored in aliquots at −80 °C until analysis. The sensitive chromogenic enzyme-linked immunosorbent assay (ELISA) (Kainos Laboratories Inc., Tokyo, Japan) was used to determine plasma ADAMTS-13:AC. The normal value of ADAMTS-13:AC was 99 ± 22%. Plasma vWF:Ag was measured via sandwich ELISA using rabbit antihuman vWF polyclonal antiserum (Dako, Glostrup, Denmark). The normal value of vWF:Ag is 102% ± 33%.

### 3.3. Statistical Analyses

All statistical data were analyzed using the GraphPad Prism version 9.0.2 for Windows (GraphPad Software, San Diego, CA, USA). Continuous variables were expressed as the mean ± standard deviation. Categorical data were analyzed using Fisher’s exact test. Continuous data were compared between groups using the unpaired *t*-test (for normally distributed data) or the Mann–Whitney *U* test (for non-normally distributed data). Areas under the receiver operating characteristic curves (AUCs) were used to evaluate the diagnostic value of ADAMTS-13:AC to identify predictive threshold values and patients with PVT. The Youden index was used to confirm the best cut-off point. To simultaneously assess the effects of multiple risk factors in survival time analyses, we used the logistic regression model. Using logistic regression analysis, factors associated with PVT in LC were identified, with variables deemed significant (*p* < 0.15) included in the subsequent multivariable logistic regression analysis. A variable had to have a *p*-value of <0.15 to be entered into the regression model [24]. The log-rank test was used to compare the survival time between the PVT and NPVT groups. A two-sided *p*-value of <0.05 was considered statistically significant.

## 4. Discussion

Approximately 30% of PVT cases in patients with LC showed improvement, and 10% achieved complete recanalization [1]. The remaining 70% comprised stable cases (45%) and progressive cases of PVT in cirrhosis. The standard treatment for PVT is not well defined, and recommendations rely on expert opinions and consensus. The ATIII agent is recommended for Japanese patients with PVT having a serum level of ATIII (≤70%). Currently, patients with PVT are treated with anticoagulant therapy, predominantly involving low-molecular-weight heparin, vitamin K antagonists, and direct oral anticoagulants (DOACs). However, the response to anticoagulant therapy in PVT is often unsatisfactory. The Baveno VII Consensus Workshop reported that patients with low platelet counts (e.g., <50 × 10^9^/L) face a higher risk of both PVT and bleeding complications on receiving anticoagulant therapy, necessitating a case-by-case evaluation [25]. Therefore, there is an urgent need for therapeutic guidelines. 

PVT affects mortality in cirrhosis. We attempted to identify diagnostic and prognostic biomarkers in patients with cirrhosis and PVT. We found a strong association between the presence of PVT in LC and low levels of ADAMTS 13:AC. An ADAMTS 13:AC level of <18.8% could serve as a predictor of PVT in patients with LC. ADAMTS-13:AC levels are inversely proportional to the severity of LC, leading to the observed imbalance between the decreased ADAMTS-13:AC level and the increased vWF:Ag level in patients with LC. ADAMTS-13 and vWF are associated with the severity of LC via hypercoagulability, which represents a significant risk factor for PVT development [26]. Regarding the characteristic background, the reason for the differences in INR, PT, and D-dimer levels between studies remains unclear. However, one possible explanation is the difference in the hepatic function reserve of patients with LC between studies. Lancellotti S et al. demonstrated that the number of patients with Child–Pugh class A LC in the non-PVT group is significantly more than that in the PVT group [14], whereas, in the current study, the hepatic function reserve of patients with LC was comparable between the PVT and non-PVT groups. Highly increased VWF:Ag levels in patients with LC contribute to the induction of primary hemostasis, resulting in PVT formation. D-dimer and fibrinogen are well-known markers of secondary hemostasis. This is the reason why no differences were found between the PVT and non-PVT groups. To the best of our knowledge, this is the first study to demonstrate that low serum ADAMTS 13 levels are significantly associated with the prognosis of patients with cirrhosis and PVT. ADAMTS 13:AC levels also serve as a diagnostic marker for PVT in LC.

Sacco M et al. demonstrated that the ADAMTS-13/VWF ratio is a reliable predictive biomarker for the development of PVT in patients in LC [16]. PVT development could be explained by microcirculatory disturbance with the formation of thrombi in a cirrhotic patient’s microvasculature induced by the imbalance between the ADAMTS-13 enzyme and the vWF substrate. vWF is a large multimeric glycoprotein produced by endothelial cells and stored in the form of ultralarge (UL) vWF multimers in Weibel–Palade bodies. The ADAMTS-13 protein is an enzyme, a vWF-cleaving protease. The imbalance between vWF and ADAMTS-13 predisposed to vWF combines with platelets to form a thrombus. Sacco et al. have recently shown that the ADAMTS 13/VWF ratio predicts the risk of developing PVT in cirrhosis [16]. Consistent with previous reports by Mikuła et al. [15] and Lancellotti et al. [14], we have shown that low ADAMTS 13 levels are independently associated with PVT in patients with LC. ADAMTS13:AC levels have better diagnostic performance with a larger AUC in this study than in previous studies [14,16]. Furthermore, ADAMTS13:AC levels show better diagnostic performance with higher sensitivity, specificity, and AUC for PVT detection than biomarkers of coagulation activation, anticoagulation, fibrinolysis, endothelial injury, and inflammation [27,28,29], indicating that ADAMTS13:AC levels are most closely associated with PVT development in patients with LC. The different etiology influences the systemic inflammatory response of patients and deeply affects ADAMTS 13 activity. The discrepancies between studies may be ascribed to the differences in the patient’s clinical features, including the etiology and hepatic function reserve.

The ADAMTS 13:AC level has previously been demonstrated to be a reliable prognostic marker for PVT in patients with LC [14,15,30]. The lack of ADAMTS-13 contributes to the accumulation of UL-VWFM released from LSECs and causes platelet-rich microvascular thrombi to develop under high shear stress, followed by sinusoidal microcirculatory disturbances. ADAMTS-13:AC levels gradually decreased with worsening hepatic function reserve in LC [31]. ADAMTS-13 activity predicts the cumulative survival of patients with LC in comparison with the CPS and the model for end-stage liver disease score (MELD score) [31]. Recombinant ADAMTS-13 administration improves the survival of patients with decompensated LC [32]. The optimal cutoff value of ADAMTS-13:AC to predict PVT was 18.8%, with the highest sum of sensitivity (97.0%) and specificity (81.8%). Consistent with our study, Lancellotti et al. demonstrated that the calculation of the AUC showed a value of 0.709 (70.9%) with a significant *p*-value (*p* = 0.005), and a cutoff ADAMTS-13 activity of 18.0% provides an ideal diagnostic value (balanced sensitivity and specificity) to predict PVT in cirrhosis with 85.9% accuracy [14]. These findings suggest that low serum ADAMTS-13:AC levels are a useful indicator of PVT in patients with LC. Thrombotic thrombocytopenia purpura (TTP), defined clinically by microangiopathic hemolytic anemia and thrombocytopenia, is attributed to either inherited or acquired loss of VWF multimer size regulation caused by severe ADAMTS13 deficiency, resulting in the accumulation of UL–VWF multimers. The perturbation of vWF proteolysis is attributable to decreased ADAMTS-13 levels, which mostly occur locally in the disturbance of hepatic microcirculation by sinusoidal thrombosis due to the altered activity of HSCs responsible for the ADAMTS-13 deficiency [14]. The imbalance of vWF/ADAMTS-13 exacerbates the low specificity of vWF proteolysis by ADAMTS-13 in patients with LC [16]. The expression of thrombogenic ultrahigh-molecular-weight vWF multimers (VWF-HMWMs) is increased in the portal venous endothelium in LC [33]. Thus, the vWF/ADAMTS-13 imbalance may facilitate a local persistence of vWF-HMWMs but not VWF levels in patients with LC and PVT [34]. Moreover, the defective secretion of ADAMTS-13 produced in HSCs causes microcirculatory thrombosis in the glomerulus, without changing the pattern of circulating VWF multimers [35]. vWF multimeric analyses are required for analyzing the concentration and distribution of vWF multimers in the serum. These findings could explain why the vWF level is not significantly different between the two groups. Thus, the ADAMTS-13 level seems to be a highly sensitive parameter for the presence of PVT in LC. These findings reinforce the central role of ADAMTS-13:AC in the pathogenesis of PVT in patients with LC and suggest that ADAMTS-13 activity exclusively serves as a diagnostic and prognostic marker in patients with LC. 

In addition, the United States Food and drug administration approved the first and only recombinant ADAMTS-13 enzyme replacement therapy for the treatment of congenital thrombotic thrombocytopenic purpura [36,37]. Recombinant ADAMTS13 provided higher ADAMTS13 exposure compared with plasma exchange in patients with acquired TTP [38]. Currently, antithrombin III is the sole recommended agent for patients with LC and PVT in Japan [39]. Future research is necessary to determine whether serum ADAMTS-13:AC levels can be used for PVT diagnosis in routine clinical practice. Furthermore, clinical trials should be conducted to evaluate the efficacy and safety of administering recombinant ADAMTS-13 to patients with PVT and ADAMTS-13:AC levels <18.8%. 

This study had several limitations. First, it was a retrospective, single-center study that involved a small number of Japanese patients with LC who either developed or did not develop PVT. Second, Sacco M et al. previously demonstrated the reliability of the ADAMTS-13/VWF ratio as a predictive biomarker for PVT development in patients with LC [16]. However, the present study did not calculate cumulative incidence rates of PVT. 

VWF multimers were not analyzed in this study. As previously explained, vWF multimeric analyses are required to analyze the concentration and distribution of vWF multimers in serum. Third, the incidence of PVT was low possibly because of the short observation period or possibly because PVT spontaneously disappeared in some patients with LC. Fourth, although we included many outpatients with LC, this study excluded those with the presence of occult thrombosis-provoking factors, those in whom CECT could not be performed, and those in whom ADAMTS13:AC and VWF:Ag could be affected. The possibility of selection bias must be mentioned because patients without comorbidities, malignancies, other thrombosis, or anticoagulation therapy were exclusively included in this study. Fifth, ADAMTS-13:AC and VWF Ag are associated with the hepatic function reserve. However, the hepatic function reserve was not significantly correlated with PVT development in the multivariate analysis because no differences in etiologies were observed between the PVT and non-PVT groups. Despite these limitations, ADAMTS-13:AC can function as a novel diagnostic and prognostic biomarker for PVT in Japanese patients with LC. However, longitudinal studies with larger sample sizes are needed to validate the robustness and reliability of ADAMTS-13:AC as a biomarker for PVT in LC. 

## 5. Conclusions 

This study reveals the relationship between serum ADAMTS-13 activity and the prognosis and diagnosis of Japanese patients with LC and PVT. These findings necessitate additional research to explore the utility of ADAMTS-13:AC as a noninvasive diagnostic and prognostic tool and a potential target for future therapeutic interventions. 

## Figures and Tables

**Figure 1 ijms-25-02678-f001:**
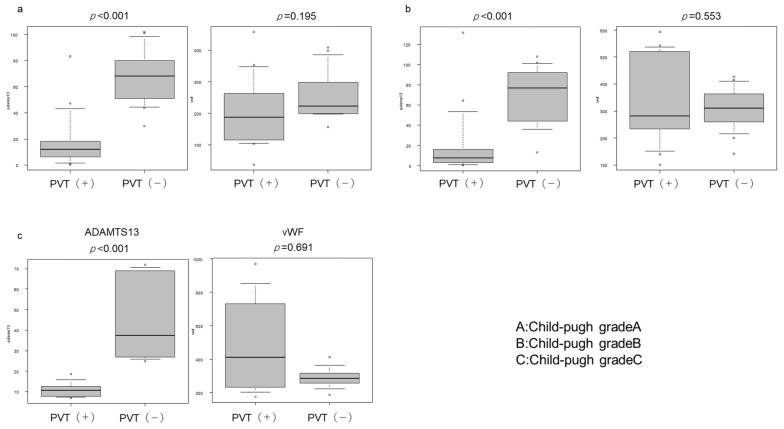
Plasma ADAMTS-13:AC and vWF:Ag levels. (**a**) Child–Pugh A (CP-A) (**b**) CP-B (**c**) CP-C, Regardless of the CP grade, ADAMTS-13:AC in patients with liver cirrhosis (LC) with portal vein thrombosis (PVT) was significantly lower than that in patients without liver cirrhosis (*p* < 0.05). No significant differences in vWF:Ag were observed between patients with LC who had PVT and those who did not have PVT (*p* < 0.05). ADAMTS-13, a disintegrin-like metalloproteinase with thrombospondin type 1 motif 13; ADAMTS-13:AC, vWF, von Willebrand factor; vWF:Ag, vWF antigen.

**Figure 2 ijms-25-02678-f002:**
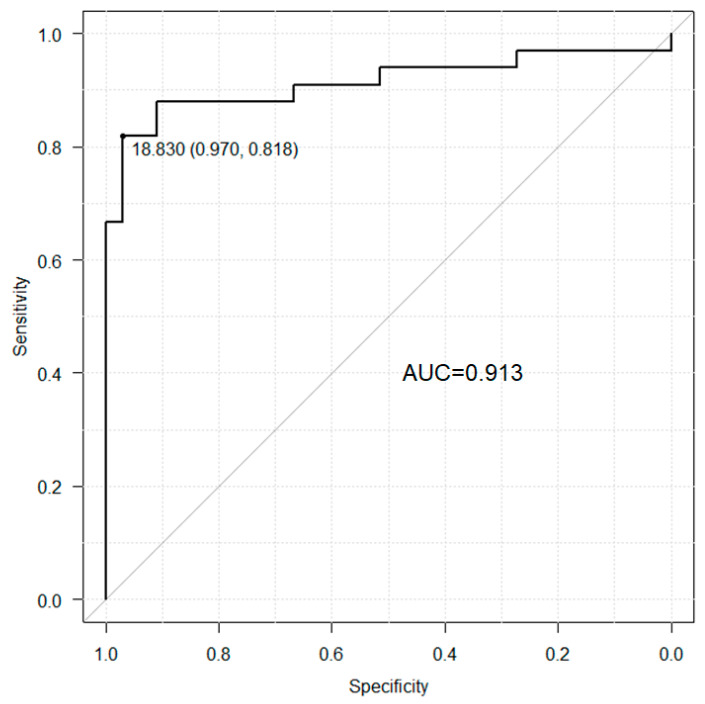
Area under the receiver operating characteristic curve of ADAMTS-13:AC for the detection of PVT in patients with cirrhosis. ADAMTS-13, a disintegrin-like metalloproteinase with thrombospondin type 1 motif 13; ADAMTS-13:AC, portal vein thrombosis; PVT.

**Figure 3 ijms-25-02678-f003:**
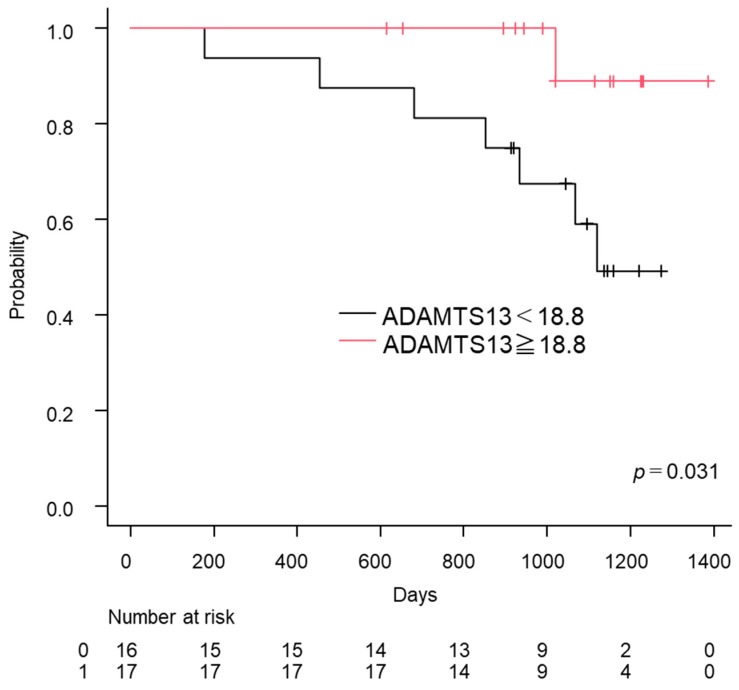
Kaplan–Meier survival curve comparisons of patients according to their optimal cutoff levels of ADAMTS-13:AC for PVT diagnosis in cirrhosis. ADAMTS-13, a disintegrin-like metalloproteinase with thrombospondin type 1 motif 13; ADAMTS-13:AC, portal vein thrombosis; PVT.

**Figure 4 ijms-25-02678-f004:**
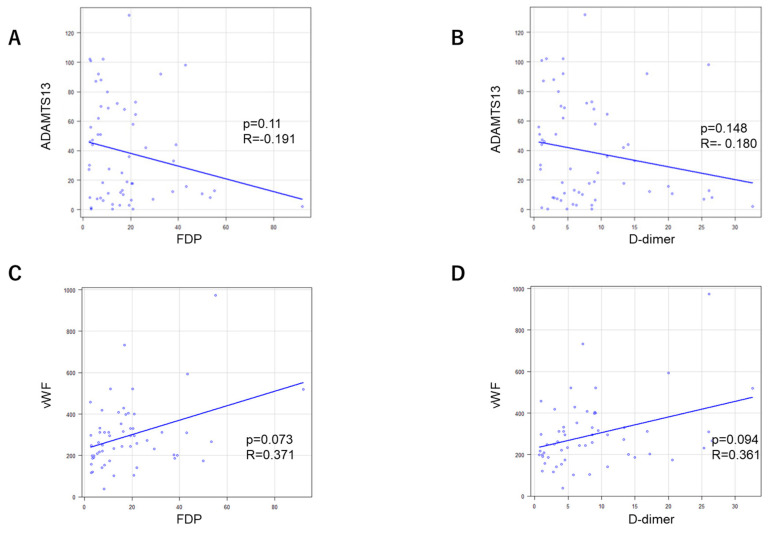
Correlations of ADAMTS-13:AC levels with either (**A**) FDP or (**B**) D-dimer levels and vWF:Ag levels with either (**C**) FDP or (**D**) D-dimer levels.

**Figure 5 ijms-25-02678-f005:**
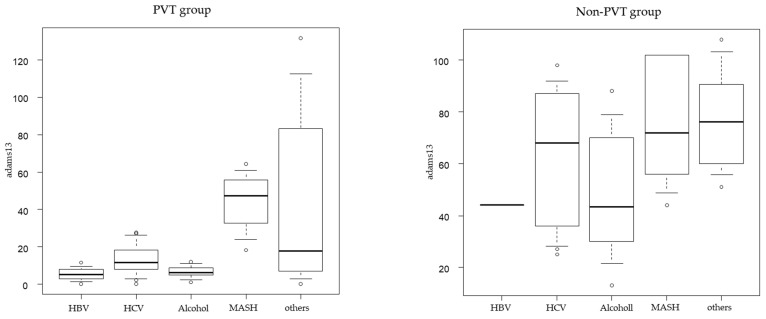
ADAMTS-13 activity levels for different etiologies.

**Table 1 ijms-25-02678-t001:** Baseline characteristics of the patients with cirrhosis.

Variables	Total (n = 66)	PVT Group (n = 33)	Non-PVT Group (n = 33)	*p* Value
Age (years)	70.0 ± 8.8	70.0 ± 9.3	70.0 ± 8.4	0.95
Sex (M/F)	40/23	20/13	20/13	1.0
Etiology (HBV/HCV/Alcohol/MASH/Others)	7/25/13/7/14	6/12/7/2/6	1/13/6/5/8	0.26
Albumin (g/dL)	3.2 ± 0.6	3.2 ± 0.55	3.3 ± 0.65	0.22
Prothrombin time (PT)-international normalized ratio	1.22 ± 0.22	1.22 ± 0.16	1.21 ± 0.27	0.89
PT (%)	72.1 ± 18	69 ± 15	74 ± 22	0.26
Total bilirubin (mg/dL)	2.1 ± 2.1	1.3 ± 1.1	3.4 ± 3.3	0.40
C-reactive protein (mg/dL)	0.65 ± 0.33	0.61 ± 0.34	0.69 ± 0.39	0.70
AST (IU/L)	53 ± 31	52 ± 37	54 ± 22	0.80
ALT (IU/L)	42 ± 37	39 ± 37	44 ± 22	0.64
γ-glutamyl transpeptidase (IU/L)	80 ± 33	78 ± 35	83 ± 35	0.69
Urea nitrogen (mg/dL)	19.3 ± 11.9	20.3 ±11.2	18.3 ± 12.7	0.52
Creatinine (mg/dL)	1.02 ± 0.90	1.19 ± 1.14	0.82 ± 0.43	0.10
Platelet (×10^4^/μL)	9.6 ± 5.5	9.5 ± 5.0	9.7 ± 6.1	0.91
α-fetoprotein (ng/mL)	5.5 ± 1.9	5.6 ± 1.9	5.5 ± 1.8	0.98
Child–Pugh grade (A/B/C)	26/28/12	13/14/6	13/14/6	1.0
Child–Purgh score	7.2 ± 1.9	7.5 ± 2.0	6.9 ± 1.9	0.27
Ascites (presence/absence)	35/31	13/20	12/21	0.80
Esophago gastric varices (presence/absence)	34/32	19/14	15/18	0.46
Transient elastography-based liver stiffness measurement (Kpa)	22.0 ± 3.9	23.5 ± 7.2	20.8 ± 5.4	0.09
Fibrinogen degradation products (μg/mL)	18.0 ± 16.7	21.4 ± 19.7	14.3 ± 11.9	0.11
D-dimer (μg/mL)	8.4 ± 7.6	9.9 ± 8.6	6.1 ± 6.0	0.10
Yerdel grade 1/2/3	28/4/1	28/4/1	-	-
ADAMTS-13:AC (%)	30.5 ± 7.2	19.8 ± 8.3	62 ± 15.3	<0.001
vWF:Ag (%)	300 ± 81	278 ± 87	323 ± 75	0.15
Survival (alive/death)	53/13	33/8	33/5	0.35
Observation period (days)	1033 ± 147	1002 ± 165	1064 ± 129	0.45

AST: aspartate transaminase; ALT: alanine transaminase; PT-INR: international normalized ratio of prothrombin time; FDP: fibrin degradation product; vWF: von Willebrand factor; ADAMTS-13; a disintegrin-like metalloproteinase with thrombospondin type 1 motifs 13; MASH: metabolic dysfunction-associated steatohepatitis.

**Table 2 ijms-25-02678-t002:** Factors involved in the development of portal vein thrombosis.

Variables	Univariate Analysis			Multivariate Analysis		
	Odds Ratio	95% Confidence Interval	*p* Value	Odds Ratio	95% Confidence Interval	*p* Value
ADAMTS-13:AC < 18.8	1.56	1.10–2.34	0.0052	1.67	1.21–3.00	0.002
Fibrin degradation product ≥ 1.0	1.030	0.992–1.07	0.119	0.97	0.723–1.89	0.81
D-dimer ≥ 4.9	1.060	0.986–1.15	0.111	1.05	0.561–1.98	0.65
Creatinine ≥ 0.72	3.140	0.6510–15.10	0.154			
Child–Pugh score ≥ 6.0	1.160	0.8930–1.51	0.266			

ADAMTS-13; a disintegrin-like metalloproteinase with thrombospondin type 1 motifs 13.

**Table 3 ijms-25-02678-t003:** Diagnostic accuracy of ADAMTS13:AC levels in portal vein thrombosis.

Model	Threshold	Sensitivity	Specificity	PPV	NPV	AUC of ROC (95% CI)
ADAMTS13:AC	18.8	0.818	0.97	0.965	0.842	0.913 (0.806-1)

PPV: positive predictive value; NPV: negative predictive value

**Table 4 ijms-25-02678-t004:** Characteristics of the patients stratified by ADAMTS-13 activity.

Variables	ADAMS13 ≥ 18.8% (n = 17)	ADAMS13 < 18.8% (n = 16)	*p* Value
Age (years)	70.8 ± 8.5	69.3 ± 6.1	0.57
Sex (M/F)	12/5	11/5	0.97
Etiology (HBV/HCV/Alcohol/NASH/Others)	3/7/3/0/4	3/5/4/2/2	0.98
Albumin (g/dL)	2.9 ± 0.6	3.3 ± 0.5	0.023
Prothrombin time (PT)-international normalized ratio	1.2 ± 0.2	1.2 ± 0.14	0.30
PT (%)	69 ± 15	70 ± 15	0.42
Total bilirubin (mg/dL)	1.5 ± 0.93	2.3 ± 1.7	0.71
C-reactive protein (mg/dL)	0.57 ± 0.35	0.72 ± 0.43	0.56
AST (IU/L)	45 ± 17	59 ± 26	0.93
ALT (IU/L)	30 ± 17	49 ± 22	0.88
γ-glutamyl transpeptidase (IU/L)	77 ± 33	79 ± 26	0.93
Urea nitrogen (mg/dL)	21 ± 15	19 ± 5.6	0.78
Creatinine (mg/dL)	1.4 ± 1.3	1.0 ± 0.3	0.33
Platelet (×10^4^/μL)	10.7 ± 5.3	8.0 ± 3.1	0.48
α-fetoprotein(ng/mL)	5.7 ± 1.9	4.8 ± 1.4	0.27
Child–Pugh grade (A/B/C)	7/6/4	6/8/2	0.54
Child–Pugh score	7.6 ± 2.2	7.3 ± 1.7	0.17
Ascites (presence/absence)	8/9	9/7	0.62
Esophago gastric varices (presence/absence)	9/8	10/6	0.57
Transient elastography-based liver stiffness measurement (Kpa)	22.7 ± 6.2	23.8 ± 5.9	0.61
FDP (μg/mL)	22 ± 16	20 ± 18	0.80
D-dimer (μg/mL)	10.1 ± 7.3	9.6 ± 9.1	0.85
Yerdel grade 1/2/3	15/2/0	13/2/1	0.52
vWF	331 ± 253	291 ± 147	0.255
Survival (alive/death)	16/1	9/7	0.011

AST: aspartate transaminase; ALT: alanine transaminase; PT-INR: international normalized ratio of prothrombin time; FDP: fibrin degradation product; vWF: von Willebrand factor; vWF: von Willebrand factor.

## Data Availability

Raw data were generated at the Nara University Hospital. The datasets generated and/or analyzed during the current study are available from the corresponding author (T.N.) upon reasonable request.

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
