# Peer review of "ADAMTS-13: A Prognostic Biomarker for Portal Vein Thrombosis in Japanese Patients with Liver Cirrhosis"

_ijms, 2024, doi:10.3390/ijms25052678_

Round 1

Reviewer 1 Report

Comments and Suggestions for Authors

Dr. Suzuki et al. presented a retrospective study enrolling 66 patients and found ADAMTS-13 might be a biomarker and prognostic biomarker for portal vein thrombosis (PVT) for patients with cirrhosis. There are some major concerns:

1.    The innovation of this study is poor. Same researches (PMID: 27220954, 35778228, 29849584) have been published with similar results as the current study.

2.    In table1, no significant difference was shown in INR, PT and D-dimer between PVT group and non-PVT group. Considering the occurrence of PVT and their hypercoagulable status, it is difficult to convince that there was no difference in INR, PT and D-dimer. In previous study (PMID: 27220954), it showed that patients in PVT group had significant increased level of D-dimer and decreased PT compared with non-PVT group. The authors should explain the reason.

3.    In table1, it showed that 5 patients died in PVT group. However, in figure3 which showed the survival of PVT group, the number of deaths is clearly more than 5. This is self-contradictory.

4.    In figure3, the authors showed that PVT patients with ADAMTS-13:AC levels < 18.8% had inferior survival and concluded that ADAMTS-13:AC levels was a prognostic biomarker for PVT patients. However, the authors did not specify whether the cause of death was related to PVT. In addition, the sample size is small indicating a significant bias. Therefore, the conclusion lacks evidence to support it.

5.    Since the authors explored the diagnostic efficacy of coagulation related factors, ADAMTS-13, patients who suffered from congenital hemostatic disorders, such as hemophilia A/B, von Willebrand disease, other inherited coagulation factor deficiency, or severe thrombocythopenia, should be excluded. But the authors did not clarify it in the method, which may lead to inaccurate research results.

6.    In discussion, the authors explained the regulation between vWF and ADAMTS-13. ADAMTS-13 can proteolyze vWF inhibiting the formation of vWF multimers and PVT. But according to the data in table1, table3 and figure1, why the vWF levels showed no significant difference when ADAMTS-13 levels decreased?

7.    At present, there is a lack of effective predictive tools or models for PVT. It is recommended to supplement the discussion with the differences and comparisons in predictive sensitivity and specificity of ADAMTS-13 and other PVT related factors or models proposed in previous studies.

Comments on the Quality of English Language

need to be improved

Author Response

Reviewer 1

Dr. Suzuki et al. presented a retrospective study enrolling 66 patients and found ADAMTS-13 might be a biomarker and prognostic biomarker for portal vein thrombosis (PVT) for patients with cirrhosis. There are some major concerns:

  1. The innovation of this study is poor. Same researches (PMID: 27220954IF: 4.6 Q2 , 35778228, 29849584) have been published with similar results as the current study.

Author response: Lancellotti S et al. (PMID: 27220954) and Sacco M et al. (PMID: 35778228) have reported that the ADAMTS13 activity (ADAMTS13:AC) and ADAMTS-13/VWF ratio, respectively, serve as PVT predictors in patients with liver cirrhosis (LC). Mikuła T et al. (PMID: 29849584) have revealed that low serum ADAMTS-13 levels can be a useful diagnostic marker of PVT in patients with decompensated LC. However, to our best knowledge, this is the first study to demonstrate that low serum ADAMTS 13 levels are significantly associated with the prognosis of patients with cirrhosis and PVT. ADAMTS 13:AC levels also serve as a diagnostic marker for PVT in LC. Furthermore, ADAMTS13:AC levels have better diagnostic performance with higher AUC in this study than in previous studies (new reference No.14 Lancellotti S et al, Intern Emerg Med 2016, new reference No.16 Sacco M et al, Dig Liver Dis 2022). ADAMTS13:AC levels show better diagnostic performance with higher sensitivity, higher specificity, and higher AUC to detect PVT than coagulation activation, anticoagulation, fibrinolysis, endothelial injury, and inflammation biomarkers (new reference No.27 and REN W et al, Clin Appl Thromb Hemost 2020, new reference No.28 Han JB et al, J Clin Transl Hepatol 2021, new reference No.29 Long Y et al, Medicine (Baltimore) 2022), indicating that ADAMTS13:AC levels are most closely associated with PVT development in patients with LC. A description of these findings has been added on page 10, lines 280-283, and page 11, lines 295-300.

  1. In table1, no significant difference was shown in INR, PT and D-dimer between PVT group and non-PVT group. Considering the occurrence of PVT and their hypercoagulable status, it is difficult to convince that there was no difference in INR, PT and D-dimer. In previous study (PMID: 27220954IF: 4.6 Q2 ), it showed that patients in PVT group had significant increased level of D-dimer and decreased PT compared with non-PVT group. The authors should explain the reason.

Author response: Regarding the characteristic background, the differences in INR, PT, and D-dimer levels between studies remain unclear. However, one possible explanation is the differences in the hepatic function reserve of patients with LC between studies. Lancellotti S et al. demonstrated that the number of patients with Child–Pugh class A LC in the non-PVT group is significantly higher than those in the PVT group (new reference No14. Lancellotti S et al, Intern Emerg Med 2016), whereas the hepatic function reserve of patients with LC in this study is comparable between the PVT and non-PVT groups. Highly increased VWF:Ag levels in patients with LC contribute to the induction of primary hemostasis resulting in PVT formation. D-dimer and fibrinogen are well-known markers for secondary hemostasis. Thus, no differences were found between the PVT and bon PVT groups. A description of these findings has been added to page 11, lines 270–283.

  1. In table1, it showed that 5 patients died in PVT group. However, in figure3 which showed the survival of PVT group, the number of deaths is clearly more than 5. This is self-contradictory.

Author response:

Table 1 shows that eight patients died in the PVT group. Table 3 shows that one patient died in the high ADAMS13 (≥18.8%) group and seven in the low ADAMS13 (<18.8%) group.

  1. In figure3, the authors showed that PVT patients with ADAMTS-13:AC levels < 18.8% had inferior survival and concluded that ADAMTS-13:AC levels was a prognostic biomarker for PVT patients. However, the authors did not specify whether the cause of death was related to PVT. In addition, the sample size is small indicating a significant bias. Therefore, the conclusion lacks evidence to support it.

Author response:

All 13 patients died of liver-related causes. ADAMTS13:AC and VWF:Ag could be affected in patients with hepatocellular carcinoma (HCC), extrahepatic malignancy, other thrombosis, and anticoagulation therapy (new reference No.21 Obermeier HL et al, Res Pract Thromb Haemost 2019). Contrast-enhanced computed tomography (CECT) was not performed in patients with severe renal dysfunction or bronchial asthma to assess PVT in patients with LC. Therefore, these patients were excluded from this study. We also added the following sentences to the Discussion section: Third, the incidence of PVT was low possibly because of the short observation period or possibly because PVT spontaneously disappeared in some patients with LC. Fourth, although we enrolled many outpatients with LC, the study excluded patients with occult thrombosis-provoking factors, those in whom CECT could not be performed, and those in whom ADAMTS13:AC and VWF:Ag could be affected. The possibility of selection bias must be mentioned because patients without comorbidities, malignancies, other thrombosis, or anticoagulation therapy were exclusively included in this study. Longitudinal studies with larger sample sizes are needed to validate the robustness and reliability of ADAMTS-13:AC as a biomarker for PVT in LC. In conclusion, this study reveals the relationship of serum ADAMTS-13 activity with the prognosis and diagnosis in Japanese patients with LC and PVT. These findings require additional research to explore the ADAMTS-13:AC utility as a noninvasive diagnostic and prognostic tool and a potential target for future therapeutic interventions. Moreover, the title of this study has been changed to tone down conclusions. A description of these findings have been added to page 3 line 140, page 3 lines 101- 104, page 12 line 352- page 13 line 361, page 13 lines 366- line 372 and page 1 lines 2-4.

  1. Since the authors explored the diagnostic efficacy of coagulation related– factors, ADAMTS-13, patients who suffered from congenital hemostatic disorders, such as hemophilia A/B, von Willebrand disease, other inherited coagulation factor deficiency, or severe thrombocythopenia, should be excluded. But the authors did not clarify it in the method, which may lead to inaccurate research results.

Author response:

No patients with congenital coagulation factor deficiencies or severe thrombocytopenia were included in this study. A description of these findings has been added to page 3, lines 110-111.

  1. In discussion, the authors explained the regulation between vWF and ADAMTS-13. ADAMTS-13 can proteolyze vWF inhibiting the formation of vWF multimers and PVT. But according to the data in table1, table3 and figure1, why the vWF levels showed no significant difference when ADAMTS-13 levels decreased?

Author response:

Thrombotic thrombocytopenia purpura (TTP) defined clinically by microangiopathic hemolytic anemia and thrombocytopenia has been known to be attributed to either inherited or acquired loss of VWF multimer size regulation caused by severe ADAMTS13 deficiency, resulting in the accumulation of UL–VWF multimers. The perturbation of vWF proteolysis is attributable to decreased ADAMTS-13 levels, which mostly occur locally in the disturbance of hepatic microcirculation by sinusoidal thrombosis as a consequence of the altered activity of HSCs responsible for the ADAMTS-13 deficiency (new reference No12. Lancellotti S et al, Intern Emerg Med 2016). The vWF/ADAMTS-13 imbalance exacerbated the low specificity of vWF proteolysis by ADAMTS-13 in patients with LC (new reference No14. Sacco M et al, Dig Liver Dis 2022). The expression of thrombogenic ultrahigh molecular weight vWF multimers (VWF-HMWMs) is increased in portal venous endothelium in LC (new reference No31. Airola C et al, Cells 2023). Thus, the vWF/ADAMTS-13 imbalance may facilitate local persistence of vWF-HMWMs, but not VWF level in patients with LC and PVT (new reference No32. Groeneveld DJ et al, J Thromb Haemost 2021). Moreover, a defective secretion of ADAMTS-13 produced in HSCs causes microcirculatory thrombosis in the glomerulus, without changing the pattern of circulating VWF multimers (new reference No33. Tati R et al, PLoS One 2011). vWF multimeric analyses are required for analyzing the concentration and distribution of vWF multimers in the serum. These findings could explain the reason why the vWF level is not significantly different between the two groups. Thus, the ADAMTS-13 level seems to be a highly sensitive parameter for the presence of PVT in LC. These findings reinforce the central role of ADAMTS-13:AC in the PVT pathogenesis in patients with LC and indicate that ADAMTS-13 activity exclusively serves as a diagnostic and prognostic marker in patients with LC. A description of these findings has been added to page 15 lines 315- 334.

  1. At present, there is a lack of effective predictive tools or models for PVT. It is recommended to supplement the discussion with the differences and comparisons in predictive sensitivity and specificity of ADAMTS-13 and other PVT related factors or models proposed in previous studies.

Author response: ADAMTS13:AC levels have better diagnostic performance with a higher AUC in this study than in previous studies (new reference No.12 Lancellotti S et al, Intern Emerg Med 2016, new reference No.14 Sacco M et al, Dig Liver Dis 2022). Furthermore, ADAMTS13:AC levels show better diagnostic performance with higher sensitivity, specificity, and AUC for PVT detection than biomarkers of coagulation activation, anticoagulation, fibrinolysis, endothelial injury, and inflammation (new reference No.25 and REN W et al, Clin Appl Thromb Hemost 2020, new reference No.26 Han JB et al, J Clin Transl Hepatol 2021, new reference No.27 Long Y et al, Medicine (Baltimore) 2022), indicating that ADAMTS13:AC levels are most closely associated with PVT development in patients with LC. A description of these findings has been added to page 14, lines 291-296.

Reviewer 2 Report

Comments and Suggestions for Authors

Suzuki et al approached a new potential biomarker for the prognosis of patients with cirrhosis to develop portal vein thrombosis. However, there are aspects that should be further addressed:

Major comments:

1.      Could the authors structure the abstract with the subtitles: introduction, material and methods, results, and conclusion for the better presentation of the article?

2.     The introduction is too short and it should be expanded.

3.      The section Materials and Methods should be placed second after the introduction. Moreover, a flowchart of the 345 excluded patients should be provided.

4.     The variables in the tables should be expressed only as mean ±standard deviation or median and IQR. Please adjust the tables. Moreover, could the authors explain why for logistic regression the variables were considered significant for p<0.15?

5.     In the Material and Methods section, the reference for the Child-Pough score should be listed in the reference, not in the text.

6.      In the results section it is stated that “ADAMTS-13:AC <18.8% is a significant risk factor for PVT in patients with LC (OR: 0.00694; 95% confidence [CI]:0.000786–0.0613; p < 0.001) (OR: 0.00684; 95% confidence [CI]: 0.000672–0.0697; p < 0.001)” . The results of the OR are too small in order to mention that there is a significant risk.

7.     For the ROC curve in section 2.3, could the authors mention the sensitivity, and specificity as long as the positive and negative predictive value? Or maybe even providing a Youden's index with the subsequent cut-off value?

8.     For the assessment of risk development PVT in patients with cirrhosis using the biomarker, is there any difference between the etiologies of the cirrhosis?

9.     The statistics must be revised.

Minor comments:

Review by a native English speaker as some phrases are oddly conceived.

Best regards,

The Reviewer

Comments on the Quality of English Language

Some corrections required

Author Response

Reviewer 2

Suzuki et al approached a new potential biomarker for the prognosis of patients with cirrhosis to develop portal vein thrombosis. However, there are aspects that should be further addressed:

Major comments:

  1. Could the authors structure the abstract with the subtitles: introduction, material and methods, results, and conclusion for the better presentation of the article?

Author response: The manuscript has been revised accordingly.

  1. The introduction is too short and it should be expanded.

Author response: The following sentences have been added to the Introduction section: The prevalence of PVT in patients with LC varied widely in reported studies, ranging from 0.65 to 15.8% (new reference No. 6 Mantaka A et al, Ann Gastroenterol 2018). The incidence of PVT was reported as 1.6% at 1 year, 6% at 3 years, and 8.3% at 5 years in a prospective study of 369 patients with LC (new reference No. 7 Turon F et al, J Hepatol 2021). PVT can spontaneously disappear during a short follow-up period (new reference No. 8 Qi X et al, BMC Med 2018). A disintegrin-like metalloproteinase with thrombospondin type-1 motifs 13 (ADAMTS13) is a zinc-containing metalloproteinase that cleaves multimeric von Willebrand factor (VWF) between Tyr1605 and Met1606 residues in the central A2 domain (new reference No.10 Yada N et al, Thromb Haemost 2024). ADAMTS13 is expressed mainly in hepatic stellate cells (HSCs) adjacent to sinusoidal endothelial cells (SECs) (new reference No.11 Uemura M et al, Blood 2005). VWF, a multimeric glycoprotein, is produced and secreted in vascular ECs of different sizes and is released into the plasma as unusually large VWF (ULvWF) multimers. The hemostatic and thrombogenic potential of vWF depends on the multimer size (new reference No.12 Crawley JT, et al, Crawley JT). VWF stimulates platelet thrombus formation in a high-shear-stress environment, and ULvWF is more likely to adhere to platelets to induce platelet adhesion and aggregation in the circulatory system (new reference No.13 Schuppner R et al, Thromb Haemost 2018). The ADAMTS-13 deficiency locally elevates ULvWF multimers in sinusoids. This phenomenon may support platelet thrombus formation within the hepatic microvasculature and macrovasculature, including the portal vein (new reference No.14 Lancellotti S et al, Intern Emerg Med 2016). Therefore, in LC, impaired ADAMTS13 production results in an elevated intra-sinusoidal presence of ULvWF multimers and plasma coagulation factors followed by PVT. A description of these findings has been added to page 3, lines 60-64 and 71-84.

  1. The section Materials and Methods should be placed second after the introduction. Moreover, a flowchart of the 345 excluded patients should be provided.

Author response: The manuscript has been modified and Figure 1 added accordingly.

  1. The variables in the tables should be expressed only as mean ±standard deviation or median and IQR. Please adjust the tables. Moreover, could the authors explain why for logistic regression the variables were considered significant for p<0.15?

Author response: A variable must have a p-value of <0.15 to be entered to the regression model (New reference No22. Liang et al, Ann Transl Med 2020). A description of these findings has been added to page 7, lines 165-166.

  1. In the Material and Methods section, the reference for the Child-Pough score should be listed in the reference, not in the text.

Author response: I listed reference No.21, Peng Y et al. Medicine (Baltimore) 2016, accordingly. A description of these findings has been added to page 5, line 132.

  1. In the results section it is stated that “ADAMTS-13:AC <18.8% is a significant risk factor for PVT in patients with LC (OR: 0.00694; 95% confidence [CI]:0.000786–0.0613; p < 0.001) (OR: 0.00684; 95% confidence [CI]: 0.000672–0.0697; p < 0.001)” . The results of the OR are too small in order to mention that there is a significant risk.

Author response: The abstract, Results section, and Table 2 have been revised accordingly.

Univariate and multivariate analyses demonstrated that ADAMTS-13:AC of <18.8% is a significant risk factor for PVT in patients with LC (OR: 1.56; 95% confidence [CI]: 1.10–2.34; p = 0.0052) (OR: 1.67; 95% confidence [CI]: 1.21–3.00; p= 0.002) (Table 2). A description of these findings has been added to page 1, lines 41-42, page 6, lines 193-195, and Table 2.

  1. For the ROC curve in section 2.3, could the authors mention the sensitivity, and specificity as long as the positive and negative predictive value? Or maybe even providing a Youden's index with the subsequent cut-off value?

Author response: Figure 2 has been revised accordingly. The cutoff predictive value for the detection of PVT was ≥18.8% with 81.8% sensitivity, 97.0% specificity, 96.5% positive predictive values, and 84.2% negative predictive values (Figure 2). The Youden index was used to confirm the best cutoff point. A description of these findings has been added to page 6, lines 193-195 and page 6 line 161.

  1. For the assessment of risk development PVT in patients with cirrhosis using the biomarker, is there any difference between the etiologies of the cirrhosis?

Author response: Figure 4 has been added as suggested. ADAMTS-13:AC levels of different etiologies were shown in the PVT and non-PVT groups. The number of patients is too small to compare ADMTS13 values among different etiologies. A description of these findings has been added to page 10 lines 244-244.

  1. The statistics must be revised.

Continuous variables are expressed as the mean ± standard deviation. Youden index was used to confirm the best cut-off point. Using logistic regression analysis, factors associated with PVT in LC were identified, with variables deemed significant (p < 0.15) included in the subsequent multivariable logistic regression analysis. A variable must have a p value less than 0.15 to be entered to the regression model (New reference No22. Liang et al, Ann Transl Med 2020). We have included a description of these findings on page 4 lines 154-168.

Minor comments:

Author response: The manuscript was reviewed by native speakers of English.

Reviewer 3 Report

Comments and Suggestions for Authors

This study explores the potential of ADAMTS-13: AC levels as diagnostic and prognostic biomarkers for portal vein thrombosis (PVT) in Japanese patients with liver cirrhosis (LC). While the study addresses an important clinical concern, several critical aspects deserve consideration:

  1. 1. The limited sample size might influence the generalizability of the findings to a broader population. Additionally, the selection criteria for patients, such as excluding those with comorbidities and malignancies, could introduce selection bias.

  2. 2. While the study identifies a correlation between lower ADAMTS-13:AC levels and PVT in LC patients, it does not delve into the underlying mechanisms or causality. Understanding why this association exists would provide valuable insights for clinicians and researchers.
  3. 3. The study does not thoroughly discuss the limitations of the research, such as the potential confounding factors that were not considered or the challenges associated with using ADAMTS-13:AC as a standalone biomarker for PVT.

  4. 4. The study's findings should be validated in larger, diverse patient populations to establish the robustness and reliability of ADAMTS-13:AC as a biomarker for PVT in LC.

In summary, while this study represents a valuable contribution to understanding PVT in LC patients and the potential role of ADAMTS-13:AC as a biomarker, it should be considered a preliminary investigation. Addressing the limitations, increasing the sample size, and providing more context regarding clinical relevance would strengthen the study's impact and its applicability in real-world medical practice.

Comments on the Quality of English Language

Minor editing of English language required.

Author Response

Reviewer 3

Review by a native English speaker as some phrases are oddly conceived.

This study explores the potential of ADAMTS-13: AC levels as diagnostic and prognostic biomarkers for portal vein thrombosis (PVT) in Japanese patients with liver cirrhosis (LC). While the study addresses an important clinical concern, several critical aspects deserve consideration:

  1. The limited sample size might influence the generalizability of the findings to a broader population. Additionally, the selection criteria for patients, such as excluding those with comorbidities and malignancies, could introduce selection bias.

Author response: I totally agree with your opinion. The prevalence of PVT in patients with LC varied widely in reported studies, ranging from 0.6% to 15.8% (New reference No. 6 Mantaka A, et al, Ann Gastroenterol 2018). The incidence of PVT was reported as 1.6% at 1 year, 6% at 3 years, and 8.3% at 5 years in a prospective study of 369 patients with LC (New reference No. 7 Turon F et al, J Hepatol 2021). PVT can spontaneously disappear during a short follow-up period (New reference No. 8 Qi X et al, BMC Med 2018). Furthermore, ADAMTS13:AC and VWF:Ag levels could be affected in patients with hepatocellular carcinoma (HCC), extrahepatic malignancy, other thrombosis, and anticoagulation therapy (new reference No21. Obermeier HL et al, Res Pract Thromb Haemost 2019). Contrast-enhanced computed tomography (CECT) could not be performed in those with severe renal dysfunction or bronchial asthma to assess PVT in patients with LC. Therefore, we exclude these patients from this study. We added the following sentences in the limitation section: VWF multimers were not analyzed in this study. As previously explained, vWF multimeric analyses are required to analyze the concentration and distribution of vWF multimers in serum. Third, the incidence of PVT was low possibly because of the short observation period or possibly because PVT spontaneously disappeared in some patients with LC. Fourth, although we enrolled many outpatients with LC, the study excluded those with occult thrombosis-provoking factors, those in whom CECT could not be performed, and those in whom ADAMTS13:AC and VWF:Ag could be affected. The possibility of selection bias must be mentioned because patients without comorbidities, malignancies, other thrombosis, or anticoagulation therapy were exclusively included in this study. Therefore, longitudinal studies with larger sample sizes are needed to validate the robustness and reliability of ADAMTS-13:AC as a biomarker for PVT in patients with LC (page 2, lines 60–63, page 3, lines 101–105, page 12, lines 352–page 13, line 358, and page 13, lines 363–365).

.

  1. While the study identifies a correlation between lower ADAMTS-13:AC levels and PVT in LC patients, it does not delve into the underlying mechanisms or causality. Understanding why this association exists would provide valuable insights for clinicians and researchers.

Author response: Thrombotic thrombocytopenia purpura (TTP) defined clinically by microangiopathic hemolytic anemia and thrombocytopenia is well known to be attributed to either inherited or acquired loss of VWF multimer size regulation caused by severe ADAMTS13 deficiency, resulting in the accumulation of UL–VWF multimers. The perturbation of vWF proteolysis is attributable to decreased ADAMTS-13 levels, which mostly occur locally in the disturbance of hepatic microcirculation by sinusoidal thrombosis as a consequence of the altered activity of HSCs responsible for the ADAMTS-13 deficiency (new reference No14. Lancellotti S et al, Intern Emerg Med 2016). The imbalance of vWF/ADAMTS-13 exacerbated the low specificity of vWF proteolysis by ADAMTS-13 in patients with LC (new reference No.16. Sacco M et al, Dig Liver Dis 2022). The expression of thrombogenic ultrahigh molecular weight vWF multimers (VWF-HMWMs) is increased in the portal venous endothelium in LC (new reference No33. Airola C et al, Cells 2023). Thus, the vWF/ADAMTS-13 imbalance may facilitate a local persistence of vWF-HMWMs, but not VWF level in patients with LC and PVT (new reference No34. Groeneveld DJ et al, J Thromb Haemost 2021). Moreover, a defective secretion of ADAMTS-13 produced in HSCs causes microcirculatory thrombosis in the glomerulus, without changing the pattern of circulating VWF multimers (new reference No35. Tati R et al, PLoS One 2011). vWF multimeric analyses are required for analyzing the concentration and distribution of vWF multimers in the serum. These findings could explain the reason why vWF:Ag levels were not significantly different between the two groups. Thus, the ADAMTS-13 level seems to be a highly sensitive parameter for the presence of PVT in LC. These findings reinforce the central role of ADAMTS-13:AC in the PVT pathogenesis in patients with LC and suggest that ADAMTS-13 activity exclusively serves as a diagnostic and prognostic marker in patients with LC. A description of these findings has been added to page 12 lines 318-338.

  1. The study does not thoroughly discuss the limitations of the research, such as the potential confounding factors that were not considered or the challenges associated with using ADAMTS-13:AC as a standalone biomarker for PVT.

Author response: We totally agree with your statement. The following sentences have been added to the Discussion section. VWF multimers were not analyzed in this study. As previously explained, vWF multimeric analyses are required to analyze the concentration and distribution of vWF multimers in serum. Third, the incidence of PVT was low possibly because of the short observation period or possibly because PVT spontaneously disappeared in some patients with LC. Fourth, although we employed many outpatients with LC, the study excluded those with the presence of occult thrombosis-provoking factors, those in whom CECT could not be performed, and those in whom ADAMTS13:AC and VWF:Ag could be affected. The possibility of selection bias must be mentioned because patients without comorbidities, malignancies, other thrombosis, or anticoagulation therapy were exclusively included in this study. Fifth, ADAMTS-13:AC and VWF Ag are associated with hepatic function reserve. However, hepatic function reserve was not significantly correlated with PVT development in the multivariate analysis because no differences in etiologies were observed between the PVT and non-PVT groups. Despite these limitations, ADAMTS-13:AC can function as a novel diagnostic and prognostic biomarker for PVT in Japanese patients with LC. A description of these findings has been added to page 12 line 352- page 13 line 363.

  1. The study's findings should be validated in larger, diverse patient populations to establish the robustness and reliability of ADAMTS-13:AC as a biomarker for PVT in LC.

Author response: I totally agree with your comment. We added the following sentences as challenges and limitations. ADAMTS13:AC and VWF:Ag levels could be affected in patients with hepatocellular carcinoma (HCC), extrahepatic malignancy, other thromboses, and anticoagulation therapy (new reference No.21 Obermeier HL et al, Res Pract Thromb Haemost 2019). Contrast-enhanced computed tomography (CECT) was not performed in those with severe renal dysfunction or bronchial asthma to assess PVT in patients with LC. Therefore, these patients were excluded from this study. Longitudinal studies with larger sample sizes are needed to validate the robustness and reliability of ADAMTS-13:AC as a biomarker for PVT in LC. This study reveals the relationship of serum ADAMTS-13 activity with the prognosis and diagnosis in Japanese patients with LC and PVT. These findings necessitate additional research to explore the utility of ADAMTS-13:AC as a noninvasive diagnostic and prognostic tool and a potential target for future therapeutic interventions. A description of these findings has been added to page 3 lines 101-104 and page 13 lines 366-372.

In summary, while this study represents a valuable contribution to understanding PVT in LC patients and the potential role of ADAMTS-13:AC as a biomarker, it should be considered a preliminary investigation. Addressing the limitations, increasing the sample size, and providing more context regarding clinical relevance would strengthen the study's impact and its applicability in real-world medical practice.

We have carefully revised the manuscript according to the Reviewer's insightful comments and provided point-by-point responses above.

Round 2

Reviewer 1 Report

Comments and Suggestions for Authors

Although the author replied, it did not solve my previous problems. For example, the lack of innovation in this study, the data in the figure and table do not match, etc.

Comments on the Quality of English Language

should be improved

Author Response

Thank you for your valuable comments. The following sentences have been added to the Discussion section. Recombinant ADAMTS13 provided higher ADAMTS13 exposure compared with plasma exchange in patients with acquired TTP. A description of these findings has been added on page 13, lines 17-18. We have gone through your comments carefully and tried our best to address them one by one. We ensure that all figures and tables represent data accurately. 

Reviewer 2 Report

Comments and Suggestions for Authors

The authors substantially addressed my previous observations, the manuscript being improved accordingly. 

Author Response

Thank you for your comments.